# Computed tomographic analysis of the dental system of three Jurassic ceratopsians and implications for the evolution of tooth replacement pattern and diet in early-diverging ceratopsians

Jinfeng Hu[1], Catherine A Forster[2], Xing Xu[3,4,5]*, Qi Zhao[4,5], Yiming He[6], Fenglu Han[1]*

[1]School of Earth Sciences, China University of Geosciences, Wuhan, China; [2]Department of Biological Sciences, The George Washington University, Washington, United States; [3]Centre for Vertebrate Evolutionary Biology, Yunnan University, Kunming, China; [4]Key Laboratory of Vertebrate Evolution and Human Origins, Institute of Vertebrate Paleontology and Paleoanthropology, Chinese Academy of Sciences, Beijing, China; [5]Center for Excellence in Life and Paleoenvironment, Chinese Academy of Sciences, Beijing, China; [6]Nanjing Museum of Paleontology, Nanjing Institute of Geology and Palaeontology, Chinese Academy of Sciences, Nanjing, China

*For correspondence:
xu.xing@ivpp.ac.cn (XX);
hanfl@cug.edu.cn (FH)

**Competing interest:** The authors declare that no competing interests exist.

**Abstract** The dental system of ceratopsids is among the most specialized structure in Dinosauria by the presence of tooth batteries and high-angled wear surfaces. However, the origin of this unique dental system is poorly understood due to a lack of relevant knowledge in early-diverging ceratopsians. Here, we study the dental system of three earliest-diverging Chinese ceratopsians: *Yinlong* and *Hualianceratops* from the early Late Jurassic of Xinjiang and *Chaoyangsaurus* from the Late Jurassic of Liaoning Province. By micro-computed tomographic analyses, our study has revealed significant new information regarding the dental system, including no more than five replacement teeth in each jaw quadrant; at most one replacement tooth in each alveolus; nearly full resorption of the functional tooth root; and occlusion with low-angled, concave wear facets. *Yinlong* displays an increase in the number of maxillary alveoli and a decrease in the number of replacement teeth during ontogeny as well as the retention of functional tooth remnants in the largest individual. *Chaoyangsaurus* and *Hualianceratops* have slightly more replacement teeth than *Yinlong*. In general, early-diverging ceratopsians display a relatively slow tooth replacement rate and likely use gastroliths to triturate foodstuffs. The difference in dietary strategy might have influenced the tooth replacement pattern in later-diverging ceratopsians.

## Editor's evaluation

This paper employs virtual analytical methods to thoroughly examine the evolution of the dental system of Jurassic ceratopsian dinosaurs. The rich dataset and level of analytical detail on tooth development and replacement patterns are accompanied by careful interpretations of functional morphology and dietary adaptation, making the paper particularly valuable for main-stream paleontologists while still appealing to the wider evolutionary biology community.

## Introduction

During the Cretaceous, the ceratopsids became one of the dominant herbivorous terrestrial clades and developed dental batteries composed of a large number of teeth that interlocked vertically and rostrocaudally in the jaw (*Edmund, 1960*; *Dodson et al., 2004*). Ceratopsids developed two-rooted teeth to facilitate vertical integration of the tooth batteries with up to 5 teeth in each vertical series (*Edmund, 1960*). This contrasts with nonceratopsid taxa such as *Protoceratops* which retain single-rooted teeth which, although compacted rostrocaudally, have no more than 2 replacement teeth in each alveolus (*Edmund, 1960*). The Early Cretaceous neoceratopsians, including *Auroraceratops* and *Archaeoceratops*, have only 1 replacement tooth in each alveolus (*Tanoue et al., 2012*). By using computed tomography, *He et al., 2018* added more detailed information on the Early Cretaceous neoceratopsian *Liaoceratops* and presented evidence of the presence of the 2 replacement teeth per alveolus and shallow sulci on the roots to facilitate close-packing. Tracts of partially resorbed functional teeth in *Liaoceratops* appear to follow the growth of the jaws. *Liaoceratops* represents the first amniote for which multiple generations of tooth remnants are documented (*He et al., 2018*).

Here, we investigate the tooth replacement pattern in even earlier-diverging Late Jurassic ceratopsians using micro-computed tomography (micro-CT) imaging. Three earliest-diverging ceratopsians were studied: *Yinlong downsi*, *Hualianceratops wucaiwanensis*, and *Chaoyangsaurus youngi* (*Zhao et al., 1999*; *Xu et al., 2006*; *Han et al., 2016*). *Yinlong* and *Hualianceratops* are from the Upper Jurassic Shishugou Formation of the Junggar Basin, Xinjiang, China (*Xu et al., 2006*; *Han et al., 2015*). *Yinlong* is one of the earliest and most complete ceratopsian dinosaurs and is known from dozens of individuals (*Han et al., 2018*), whereas *Hualianceratops* is known from only the holotype, a partial skull and mandible (*Han et al., 2015*). *Chaoyangsaurus* is from the Upper Jurassic Tuchengzi Formation of Liaoning Province, China, and is represented by a partial skull and paired mandibles (*Zhao et al., 1999*). This study provides crucial new evidence in our understanding of the initial evolution of ceratopsian dental specializations and diet.

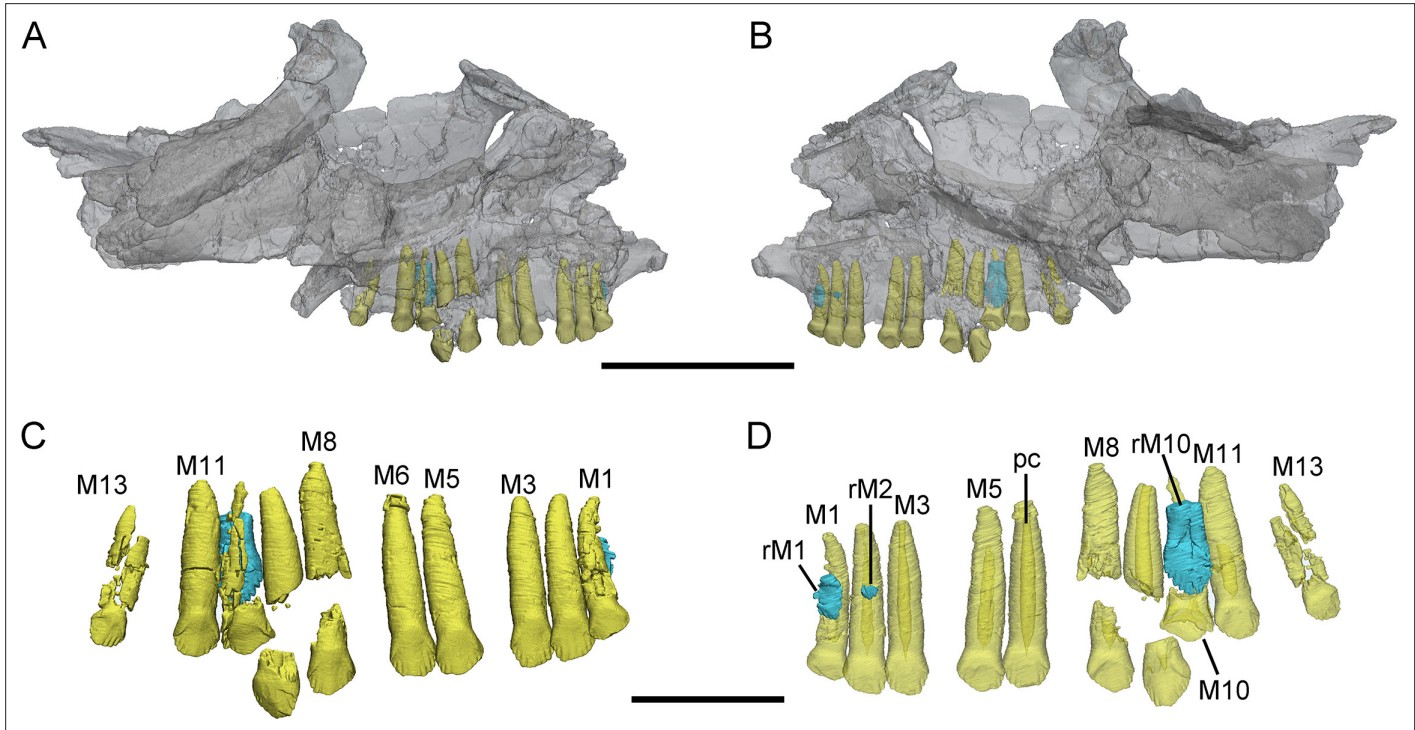

**Figure 1.** 3D reconstructions of maxillary teeth in *Yinlong downsi* (IVPP V18638). Transparent reconstructions of the right maxilla in labial (**A**) and lingual (**B**) views, and right maxillary dentitions in labial (**C**) and lingual (**D**) views. The reconstructions of maxillary dentitions are transparent in D. Elements in the CT reconstructions are color coded as follows: functional maxillary teeth, yellow; replacement teeth, cyan. Abbreviations: M1–M13, the 1st to 13th functional teeth in the maxilla; rM1, rM2, and rM10, the replacement teeth in the 1st, 2nd, and 10th tooth alveolus; pc, pulp cavity. Scale bars equal 5 cm (**A, B**) and 2 cm (**C, D**).

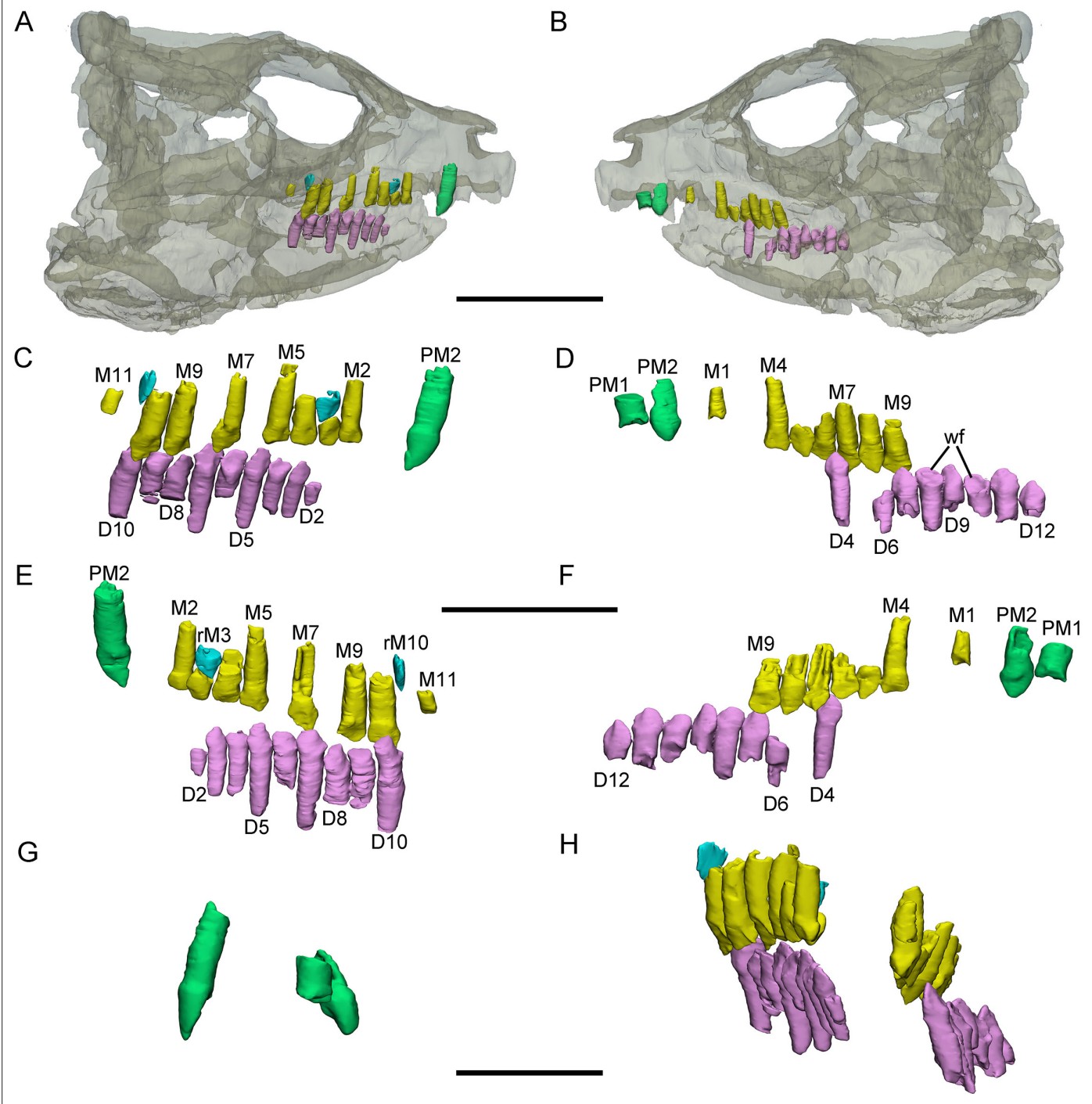

**Figure 2.** 3D reconstructions of premaxillary and cheek teeth in *Yinlong downsi* (IVPP V18636). Transparent reconstruction of the skull in right (**A**) and left (**B**) lateral views. The right tooth rows in labial (**C**) and lingual (**E**) views. The left tooth rows in labial (**D**) and lingual (**F**) views. The premaxillary teeth in rostral (**G**) view. Maxillary and dentary dentitions in rostral (**H**) view. Elements in the CT reconstructions are color coded as follows: functional premaxillary teeth, green; functional maxillary teeth, yellow; functional dentary teeth, lavender; replacement teeth, cyan. Abbreviations: M1–M11, the 1st to 11th functional teeth in the maxilla; rM3 and rM10, the replacement teeth in the 3rd and 10th alveolus; D2–D12, the 2nd to 12th functional teeth in the dentary; PM1 and PM2, the 1st and 2nd premaxillary functional teeth; wf, wear surface. Scale bars equal 5 cm (**A, B**), 3 cm (**C–F**), and 2 cm (**G, H**).

## Results

### Dentition of the early-diverging ceratopsian *Yinlong*

Premaxillary teeth. IVPP V18638 only preserves the right maxilla (*Figure 1*). All premaxillae bear 3 alveoli (*Figures 2–4*), and all three teeth are preserved in IVPP V14530 (*Figure 3C*). In IVPP V18636, the rostral two functional teeth are preserved in the left premaxilla and the second functional tooth is shown in the right premaxilla (*Figure 2C, D*). In the largest specimen (IVPP V18637), the second left functional premaxillary tooth has been lost and a replacement tooth remains in the alveolus (*Figure 4E, G*). The right premaxilla is incomplete and the first tooth is slightly damaged and the second and third are only present with roots (*Figure 4D, F*).

The digital reconstructions show that the second functional premaxillary tooth is larger than all maxillary or dentary teeth, and the third premaxillary tooth crown is quite short (*Figure 3A, C, H*). The labial surface of the premaxillary teeth is convex (*Figures 2G, 3G* and *4B*). Compared with the functional teeth on the maxilla, the long axes of the roots of the premaxillary teeth incline more dorso-lingually (*Figures 2G, H*, *3E* and *4B*).

All well-developed roots of the functional teeth in the premaxilla are nearly conical and compressed labiolingually into an oval cross-section. Compared with other premaxillary teeth, the tip and root of the second premaxillary tooth curve more distally to appear arched in lateral view (*Figures 2C and 3C,H*). The functional crowns in the premaxilla are semiconical in shape and have similar rhomboidal outlines in lateral view. They taper apically without excessive wear (*Figures 2D*, *3D, H*, *4E and G*). In rostral view, the crowns are slightly compressed labiolingually, with the lingual surface flattened and the labial rounded (*Figures 2G and 3G*). The crown morphology of the first and second premaxillary teeth in IVPP V14530 is slightly different, with an abrupt step in the second premaxillary tooth between the inflated base and the lingually flattened crown above (*Figure 3G*; *Xu et al., 2006*). The crown of the second tooth in IVPP V18636 also possesses this step but is more weakly developed than in IVPP V14530 (*Figure 2E, G*).

Premaxillary replacement teeth are only preserved in the largest skull (IVPP V18637) (*Figure 4E, G*). In lingual view, replacement teeth are present in the first and second alveoli of the left premaxilla (*Figure 4G*). They are positioned lingual to their corresponding functional teeth although the functional tooth in the second alveolus is missing. The rootless replacement tooth in the first alveolus lies adjacent to the lingual wall of the functional tooth root. The apex of its crown is positioned halfway down the root of its functional tooth (*Figure 4G*). Slight resorption can be seen in the lingual side of the root of the left first functional tooth (*Figure 4E, G*). The cross-section shows that the pulp cavity in the first replacement tooth is larger than that of the functional tooth, with a thinner layer of dentine. The apex of the first replacement tooth is more acuminate than that of the corresponding functional tooth (*Figure 4G*). The first replacement tooth is nearly triangular in lingual and labial view with an oval, mesiodistally elongated, and labiolingually compressed cross-section (*Figure 4G*). The second replacement tooth in the left premaxilla is newly erupted and only preserves the tip of the crown. The replacement premaxillary teeth in *Liaoceratops* have cone-shaped crowns and are similar in morphology to their corresponding functional teeth (*He et al., 2018*). In *Liaoceratops*, 1 or 2 replacement teeth exist in each premaxillary alveolus.

Maxillary teeth. The incomplete right maxilla of IVPP V18638 contains 10 functional teeth and 3 empty alveoli (*Figure 1A*). The left and right maxillae of IVPP V18636 contain 7 functional teeth and 8 functional teeth, respectively, with some empty sockets (*Figure 2C, D*). According to cross-sections, 4 empty sockets in the left maxilla and 3 empty sockets in the right maxilla can be discerned in IVPP V18636. Both the left and right maxillae of the holotype contain 13 functional teeth as identified before (*Figure 3C, E*; *Xu et al., 2006*; *Han et al., 2018*). However, in the largest specimen IVPP V18637, the incomplete maxillae contain 7 functional teeth and 14 functional teeth on the left and right sides, respectively (*Figure 4D, E*). The left maxilla of IVPP V18637 contains 7 empty sockets, suggesting that the maxilla bears 14 or more teeth in an adult *Yinlong*.

The maxillary tooth row is curved lingually (*Figures 3E and 4B*). Generally, the length of functional teeth increases to a maximum in the middle part of the maxillary tooth row and then decreases caudally (*Figures 1–4*). All roots of functional teeth are widest at their crown bases and taper apically to form elongated roots with a subcircular cross-section (*Figures 1C, 3H and 4D*). The root cross-sections reveal a pulp cavity surrounded by a thick layer of dentine. According to our 3D reconstructions and cross-sections, the pulp cavities of some functional teeth are open at their tips such as M3

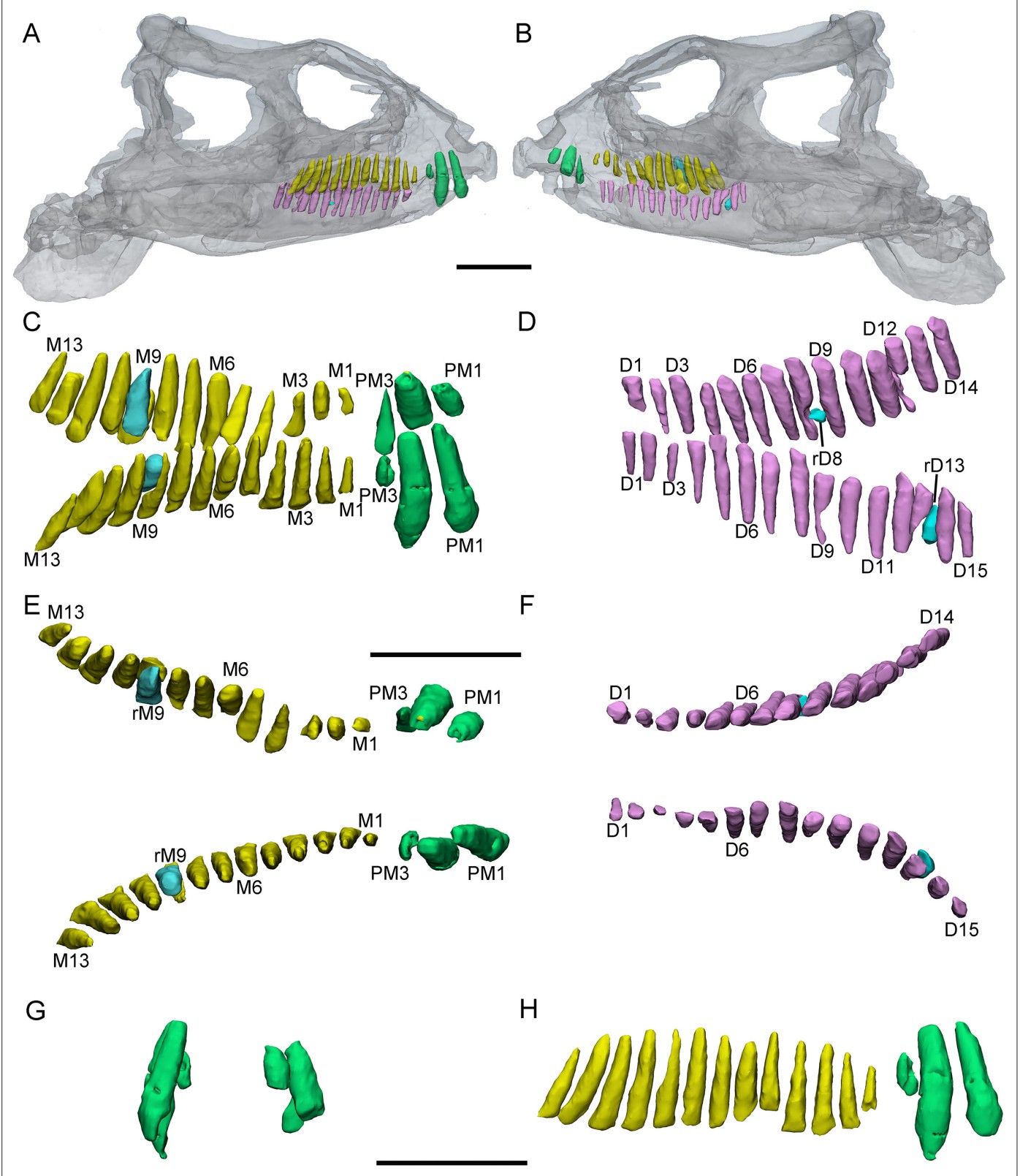

**Figure 3.** 3D reconstructions of premaxillary and cheek teeth in *Yinlong downsi* (IVPP V14530). Transparent reconstructions of the skull in right (**A**) and left (**B**) lateral views. The premaxillary and maxillary dentitions in labiodorsal (**C**) view. The dentary dentitions in labiodorsal (**D**) view. Tooth rows in the upper (**E**) and lower (**F**) jaws in dorsal view. The premaxillary teeth in rostral (**G**) view. The right tooth row in the upper jaw in labial (**H**) view. Elements in the CT reconstructions are color coded as *Figure 2*. Abbreviations: M1–M13, the 1st to 13th functional teeth in the maxilla; rM9, the replacement tooth

*Figure 3 continued on next page*

*Figure 3 continued*

in the 9th alveolus; D1–D15, the 1st to 15th functional teeth in the dentary; PM1–PM3, the 1st to 3rd functional teeth in the premaxilla; rD8 and rD13, the replacement teeth in the 8th and 13th alveolus. Scale bars equal 5 cm (**A, B**) and 4 cm (**C–H**).

and M9 in IVPP V18638 and the functional teeth with the open pulp cavity have a thinner layer of dentine (*Figure 1D*). The elongated pulp cavity in the functional tooth nearly extends over the whole root (*Figure 1D*). In all specimens, strong root resorption is seen on the lingual surface of some functional teeth adjacent to replacement teeth (*Figures 1D*, *2F*, *3C and D*). In these cases, the dentine has been resorbed by the replacement teeth such that the root base has been hollowed (*Figure 3C*). The root of M4 on the right maxilla of the holotype is also hollowed, but no replacement tooth is present (*Figure 3C*). M4s are hollowed less than D8 which is attached by a replacement crown tip. Therefore, M4 may represent the primary stage of the resorption prior to replacement tooth development.

The crowns of functional teeth in the maxilla have a spatulate outline in labial view and are slightly bulbous at the base (*Figures 1C*, *2C, D*, *3H*, *4E and F*). In IVPP V18638, all of the crowns are relatively complete with the apex of most of the crowns (except M1 and M10) showing slight wear (*Figure 1C*). The mesiodistal length and labiolingual width of erupted crowns increase to their base. In labial view, several denticles are distributed over the margin beneath the base of the crown (*Figure 1C*). Approximately 4 denticles are distributed over the mesial and distal carinae of tooth crowns and all the denticles are subequal in size and taper apically (*Figure 1C*). This feature is present but weakly developed in *Chaoyangsaurus*, *Psittacosaurus*, *Liaoceratops*, and *Archaeoceratops* (*Sues et al., 2009*). The primary ridge is prominent in M13 of IVPP V18638 and centered on the crown. The lingual surfaces of crowns are concave except for M9 whose lingual surface is convex (*Figure 1D*). In addition, M10, which is in the replacement process, has a more concave lingual surface than other functional teeth that have not undergone resorption. Therefore, we hypothesize that the lingual surfaces of the crowns are flat and gradually become concave as the wear facet develops (*Figure 1D*). Similar wear facets can be seen in *Heterodontosaurus tucki* (*Sereno, 2012*).

The count of the replacement teeth in the maxilla of *Yinlong* is 1 out of 13 functional teeth in the holotype. The smallest specimen (IVPP V18638) has the most replacement teeth in the maxilla and CT data reveal 3 replacement teeth out of 10 functional teeth inside the right maxilla (*Figure 1D*). The replacement tooth (rM10) in the holotype occurs lingual to M10 whose root has been almost completely resorbed with only a fragmented layer of dentine remaining. This replacement tooth is well developed and consists of the complete crown and partial root. The apex of rM10 reaches the base of the crown of the functional tooth. Compared with the functional teeth, the crowns of the replacement teeth are rhomboidal in labiolingual view, compressed labiolingually, and the denticles extend along nearly the entire margin of the crown (*Figure 1D*). In IVPP V18636, there are 2 replacement teeth preserved in the right maxilla (*Figure 2C, E*). The first replacement tooth, preserving only the crown, is attached to the lingual side of M3. The base of the corresponding functional tooth has been hollowed and the root has been resorbed although the crown is still functional (*Figure 2C*). In IVPP V18636, the crown of rM10 is positioned distal to M10 and is similar to the premaxillary replacement tooth of IVPP V18637 in having a triangular outline in labiolingual view (*Figures 2E and 4G*). This suggests that a replacement tooth with a labiolingually compressed shape is relatively common in *Yinlong*.

Remnants of resorbed functional teeth occur in IVPP V18637. The remnants are positioned labiodistal to functional M11 and M14 in the right maxilla (*Figure 4A, D*). Remnants of resorbed functional teeth preserve a thin layer of dentine and exhibit a crescent outline in cross-section. There is only 1 generation of resorbed tooth remnants along the maxillary tooth row. Remnants of resorbed functional teeth are also reported in *Liaoceratops*, *Coelophysis*, and a hadrosaurid, but the number of resorbed functional teeth in *Liaoceratops* is far greater than in *Yinlong* (*Bramble et al., 2017*; *LeBlanc et al., 2017*; *He et al., 2018*). In the holotype of *Liaoceratops*, about 28 remnants of the functional teeth are preserved in the right maxilla and at most 4 generations of teeth remnants are located at the middle part of the tooth row.

Dentary teeth. The holotype has a complete dentary containing 15 functional teeth on the left and 14 functional teeth on the right (*Figure 3D, F*). The dentaries of IVPP V18636 are incomplete, containing 9 functional teeth and 1 empty socket on the right dentary and 8 functional teeth and 3 empty sockets on the left (*Figure 2C, D*). The left dentary is distorted so that the long axes of

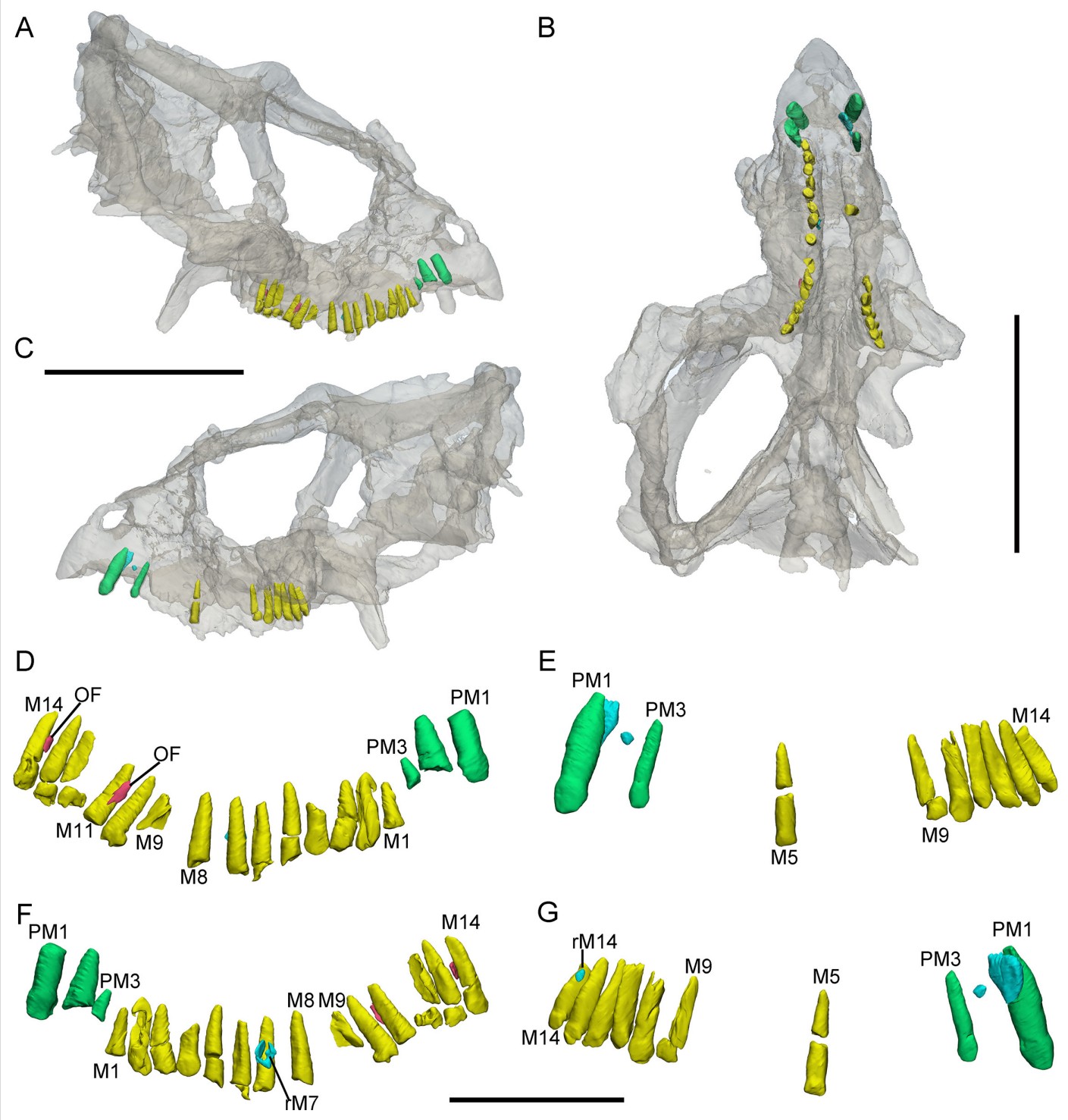

**Figure 4.** 3D reconstructions of premaxillary and maxillary teeth in the largest specimen (IVPP V18637) of *Yinlong downsi*. Transparent reconstructions of the skull in right (**A**), occlusal (**B**), and left (**C**) views. Right tooth row in labial (**D**) and lingual (**F**) views. Left tooth row in labial (**E**) and lingual (**G**) views. Elements in the CT reconstructions are color coded as *Figure 2* and remnants of functional teeth are coded as red. Abbreviations: OF, remnants of the old functional tooth; M1–M14, the 1st to 14th functional teeth in the maxilla; PM1–PM3, the 1st to 3rd functional teeth in the premaxilla; rM7, the replacement tooth in the 7th alveolus. Scale bars equal 10 cm (**A–C**) and 4 cm (**D–G**).

functional teeth on the two sides extend in different directions (*Figure 3H*). The size of the dentary teeth increases to a maximum at tooth 5 and 6 and then decreases caudally in the jaw.

The morphologies of dentary roots are similar to those of the maxillary teeth with a nearly conical shape and oval, labiolingually compressed cross-sections (*Figures 2E, F and 3D*). Most functional teeth in the dentary have complete crowns (*Figure 2C, D*). In labial view, the outline of functional teeth in the dentary is similar to maxillary teeth but its labial surface is concave (*Figure 2D*). This concave surface has never been found in other ceratopsians, suggesting that *Yinlong* had relatively precise occlusion.

Two replacement teeth can be seen in the dentary of the holotype (*Figure 3D*). Among them, the roots of D9 in the left dentary and D12 in the right dentary have been hollowed although no replacement tooth is preserved. However, the cavity caused by the resorption is similar to its corresponding functional tooth on the contralateral side (*Figure 3D, F*). It can be concluded that they are at a similar stage of the replacement process. In addition, D8 on the right side also exists a replacement tooth. Patterns of symmetry in replacement patterns can be seen in *Yinlong*, but the replacement stage between two dentaries is slightly different.

## Dentition of *Chaoyangsaurus*

The holotype of *Chaoyangsaurus* (IGCAGS V371) preserves the 2 premaxillary teeth. The premaxillary teeth of *Chaoyangsaurus* are ellipsoidal in cross-sections and the crowns are not expanded mesiodistally at their base as in *Yinlong.* The apices of the crowns are missing (*Figure 5E–H*). The first functional tooth in the left premaxilla is undergoing replacement and its corresponding replacement tooth crown is triangular with the apex inclined distally (*Figure 5E*). The long axis of the replacement tooth in the premaxilla retains the same angle of tilt with its corresponding functional tooth (*Figure 5E*).

CT reconstructions reveal that the maxillary teeth of *Chaoyangsaurus* possess different crown morphology from *Yinlong*. In *Chaoyangsaurus*, the primary ridges are located more distally on the teeth (*Figure 5F, H*) and the basal ridge extends over more than 70% of the crown with denticles spread over the mesial and distal margins (*Figure 5F*). The lingual surfaces of the maxillary crowns are concave and the crowns in the dentary also show concave surfaces similar to the situation in *Yinlong* (*Figure 5E, I and J*). The concave surface in the lingual side of maxillary crowns and the labial side of dentary crowns may indicate wear facets similar to those of *Yinlong*. The roots of the teeth in *Chaoyangsaurus* are elongated and inclined lingually. CT data also reveal the phenomenon that the fourth and seventh functional teeth have pulp cavities open at their tip and these teeth show less wear than others (*Figure 5F, H*). Therefore, the functional teeth with open pulp cavities may be newly erupted.

The morphology of the dentary teeth is similar to that of maxillary teeth although no primary ridges or denticles exist on the dentary crowns (*Figure 5I, J*). The left dentary of *Chaoyangsaurus* possesses 3 replacement teeth out of 9 functional teeth and on the other side there are 5 replacement teeth out of 11 functional teeth (*Figure 5I, J*). According to 3D reconstructions of maxillary and dentary teeth, the pulp cavity is gradually reduced through time after tooth eruption (*Figure 5F, H*). The number of replacement teeth in *Chaoyangsaurus* is slightly more than that of *Yinlong*.

## Dentition of *Hualianceratops*

The crowns of the teeth in *Hualianceratops* are similar to those of *Yinlong* but differ in some respects (*Figure 6C, D*). The dentary preserves the complete morphology of the crowns. They are subtriangular in labiolingual view and the mesial and distal margins bear about 7 denticles, respectively (*Figure 6C*), more than in *Yinlong*. Ten functional dentary teeth are identified. The tooth crowns are slightly imbricated with the distal margin of each tooth overlapping the lingual side of the mesial margin of the preceding tooth. The first functional tooth is broken with only part of the root remaining. Five replacement teeth are exposed on the lingual aspect of their corresponding functional teeth and exposed at the border of the alveoli (*Figure 6D*).

## Replacement progress and tooth development in *Yinlong* and *Chaoyangsaurus*

In *Yinlong* and *Chaoyangsaurus*, the resorption of the functional tooth is initiated before the successional tooth has germinated (*Figures 1D and 5G, I*). The functional tooth roots are resorbed resulting in a depression on the middle part of the roots (*Figures 1D and 5I*). After the depression extends

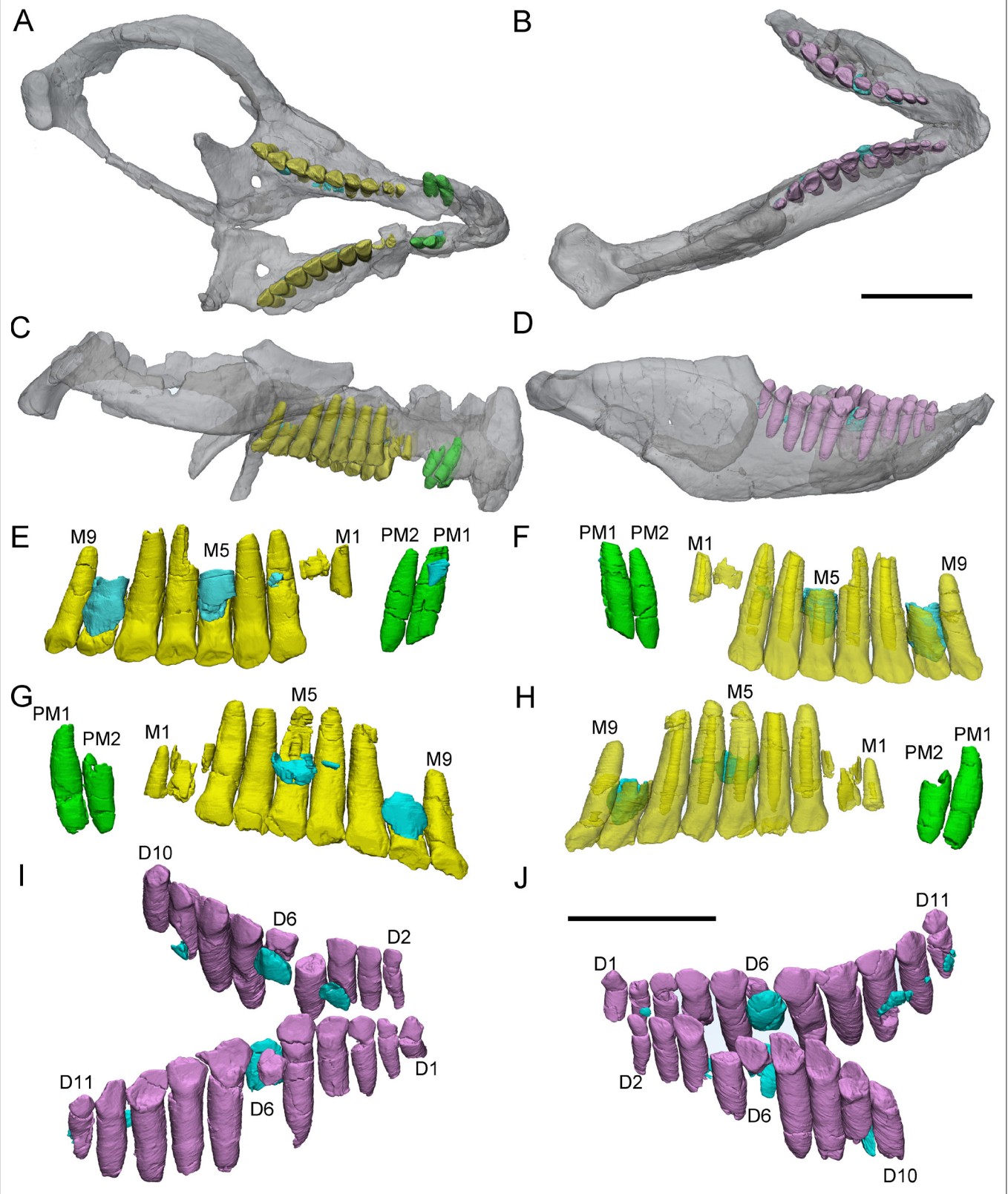

**Figure 5.** 3D reconstructions of premaxillary and cheek teeth in *Chaoyangsaurus youngi* (IGCAGS V371). Transparent reconstructions of the skull in occlusal (**A**) and right lateral (**C**) views. Transparent reconstructions of the mandible in occlusal (**B**) and right lateral (**D**) views. Left maxillary and premaxillary dentitions in lingual (**E**) and labial (**F**) views. Right premaxillary and maxillary dentitions in lingual (**G**) and labial (**H**) view. Dentary dentitions in right dorsal (**I**) and left dorsal (**J**) views. The reconstructions of maxillary dentitions are transparent in F and H. Elements in the CT reconstructions are

*Figure 5 continued on next page*

*Figure 5 continued*

color coded as *Figure 2*. Abbreviations: M1–M9, the 1st to 9th functional teeth in the maxilla; D1–D11, the 1st to 11th functional teeth in the dentary; PM1–PM2, the 1st to 2nd functional teeth in the premaxilla. Scale bars equal 3 cm (**A–D**) and 2 cm (**E–J**).

enough, the replacement teeth form lingual to the functional tooth roots with the crown situated a small distance away from the middle part of the roots. The replacement tooth crown then gradually grows crownward toward the margin of the alveolus. The most immature replacement teeth are represented by small cusps (*Figures 1D and 5E, G* ). With ontogeny, the crowns of more mature teeth become fully developed and largely resorb the lingual aspects of the roots of the functional

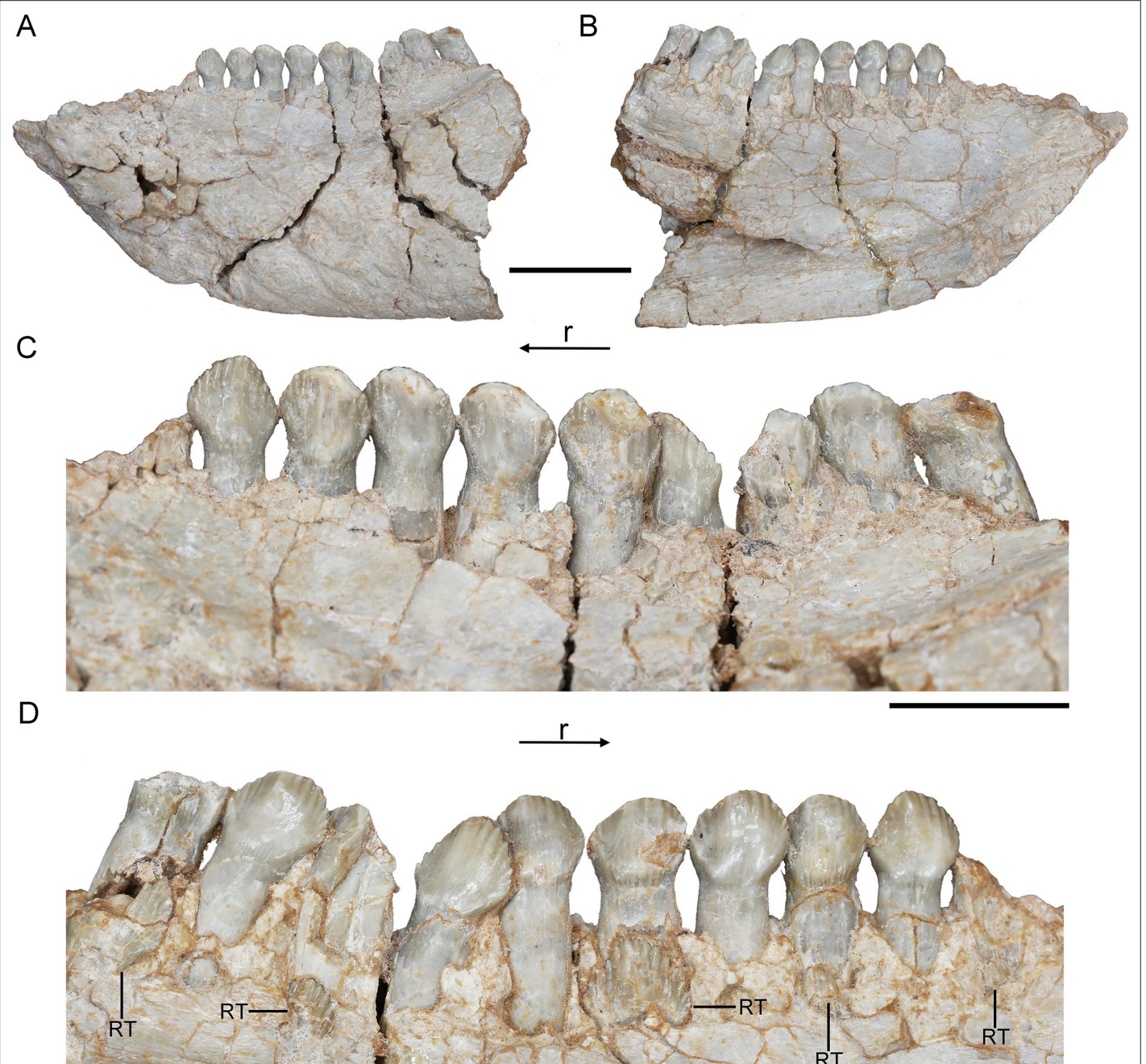

**Figure 6.** *Hualianceratops wucaiwanensis*, IVPP V28614. The left dentary in labial (**A**) and lingual (**B**) views. Dentary tooth row in labial (**C**) and lingual (**D**) views. Abbreviations: r, rostral; RT, replacement tooth. Scale bars equal 2 cm (**A, B**) and 1 cm (**C, D**).

teeth, and become partially housed in their pulp cavities (*Figures 1C*, *3D*, *4G*, *5G*). In this stage, some replacement tooth crowns in *Yinlong* and *Hualianceratops* were flat labiolingually and possibly kept this morphology until erupted (*Figures 1D, 2C, 4G and 6D*). However, the replacement crowns in *Chaoyangsaurus* were inflated and the morphology was almost the same as that of the functional teeth (*Figure 5E,G*). Differing from the maxillary teeth, the crowns of the premaxillary replacement teeth are housed in the more apical part of the functional tooth root in *Yinlong* and a similar situation occurs in *Chaoyangsaurus* (*Figures 4E, G and 5E*). As the lingual surface of the functional teeth becomes heavily resorbed, the replacement teeth reach about 60% or more of their predicted full size (*Figure 1D*). When the replacement tooth grows to its final size, most of the roots of the predecessors have faded through heavy resorption and may leave small root remnants on the labial surface of its successor's tooth (*Figure 7A, B, D, E*).

## The Zahnreihen in *Yinlong* and *Chaoyangsaurus*

In the Zahnreihen graph of IVPP V18638, these teeth show the regular pattern that the growth stage decreases progressively over a two- or three-tooth position period and hence at least 4 Zahnreihen are possibly identified (*Figure 8B*). The resulting Zahnreihen are formed by M1 to M3, M5 to M6, M8 to rM10, and M10 to M11, respectively, and run more or less parallel to each other (*Figure 8B*). The M1–M3 and M8–M10 are well-defined tooth replacement series and the exceptions are rM1, rM2, and M13. In *Yinlong*, Z-spacing is between 1.5 and 3.0, and the average *Z*-spacing is 2.54. In *Chaoyangsaurus*, *Z*-spacing is 2.0 and 3.33 with an average of 2.67. *Edmund, 1960* suggested that the *Z*-spacing in reptilian dentitions is higher in the rostral region of the tooth row generally. This pattern is also present in *Yinlong*, whereas *Z*-spacing is higher in the caudal region of the tooth row in *Liaoceratops* (*He et al., 2018*). *Fastnacht, 2008* suggested that the replacement ratio of tooth formation against tooth resorption can be directly derived by the *Z*-spacing. The replacement ratio represents the replacement rate to a certain extent but is only comparable within a single taxon. The lower the value is, the higher the tooth replacement rate (*Fastnacht, 2008*). Therefore, *Z*-spacing provides an index to compare the replacement rate in one taxon or jaw element.

The lower *Z*-spacing in the caudal maxillary region of *Yinlong* may suggest that this region of the tooth row has a higher replacement rate. To maintain the efficiency of chewing, it is advantageous to replace more rapidly worn teeth at a higher rate. Therefore, this may indicate that the caudal region of the jaw in *Yinlong* is used more than the rostral portion to chew food. The situation in *Liaoceratops* and *Chaoyangsaurus* is the opposite of that in *Yinlong* in that the rostral jaw region has a higher replacement rate and the food preparation may therefore occur more frequently in that region (*Figure 8C–F*; *He et al., 2018*). This suggests that there may be a transfer of the position of the main chewing region during the evolution of early-diverging ceratopsians.

*DeMar, 1972* reported that the value of the *Z*-spacing ranges from 1.56 to 2.80 in most reptiles. *Z*-Spacing as the quantitative index could be used to assess the replacement patterns and avoid arbitrary interpretation of replacement patterns and facilitates objective comparison of patterns between different jaw elements, individuals, growth stages, taxa, and so forth (*Hanai and Tsuihiji, 2019*). In *Liaoceratops*, the spacing between Zahnreihe ranges from 2.16 to 2.90 with a mean value of 2.58 (*He et al., 2018*). So far, only the *Z*-spacings of *Yinlong*, *Chaoyangsaurus*, and *Liaoceratops* are known in ceratopsians and more research on the *Z*-spacing of ceratopsians are required to make meaningful comparisons. In non-avian dinosaurs, all known *Z*-spacing values are greater than 2.0 (*Chatterjee and Zheng, 2002*; *Weishampel et al., 2004*; *Wiersma and Sander, 2016*; *Hanai and Tsuihiji, 2019*; *Becerra et al., 2020*). *Hanai and Tsuihiji, 2019* examined some extant crocodiles such as *Alligator mississippiensis* and *Crocodylus siamensis* which present infrequent *Z*-spacing less than 2.0. These values indicate the replacement wave direction which is rostral to caudal when *Z*-spacing is greater than 2.0, reversed when less than 2.0 and replaced in simple alternation between odd- and even-numbered tooth positions when exactly 2.0 (*Hanai and Tsuihiji, 2019*). This indicates that new teeth erupt from caudal to rostral order in either odd- and even-numbered alveoli in the maxilla of *Yinlong* and *Chaoyangsaurus*.

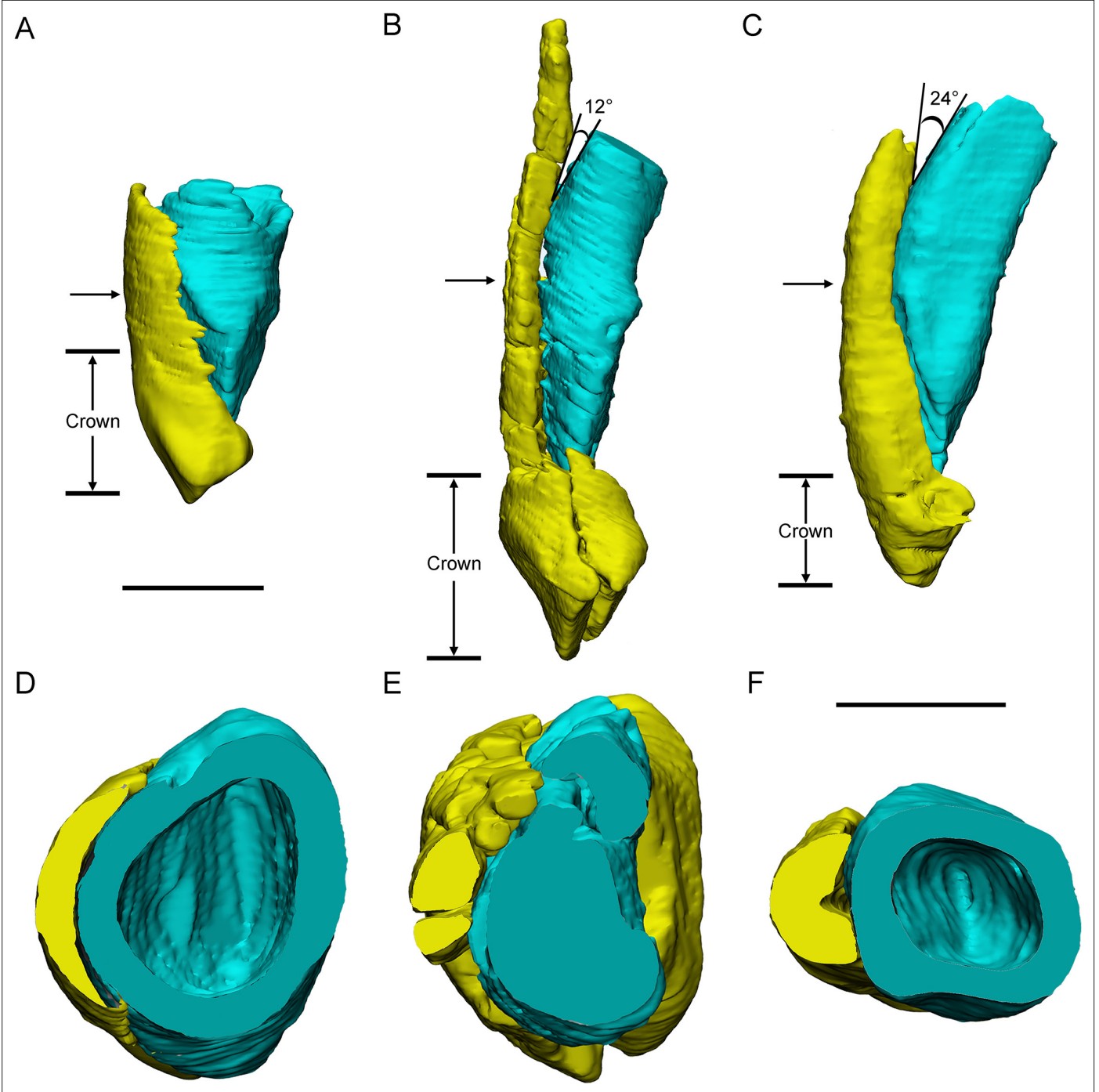

**Figure 7.** Three different replacement processes illustrated by teeth at similar replacement stage of *Chaoyangsaurus* (**A, D**), *Yinlong* (**B, E**), and *Liaoceratops* (**C, F**). The tooth 8 in the left maxilla of IGCAGS V371 in distal view (**A**) and cross-section (**D**). The tooth 10 of IVPP V18638 in mesial view (**B**) and cross-section (**E**). The tooth 7 in the right maxilla of the holotype of *Liaoceratops* (IVPP V12738) in mesial view (**C**) and cross-section (**F**). Elements in the CT reconstructions are color coded as *Figure 2*. The arrows of A–C indicate where the cross-sections generate. The replacement teeth here have developed the complete crown and part of the root. The root of the replacement tooth in *Liaoceratops* inclines lingually and that in *Yinlong* also inclines lingually but with a smaller angle of inclination. The root of the replacement tooth in *Chaoyangsaurus* clings to its corresponding functional tooth tightly. The resorbed area on the functional tooth is larger in *Chaoyangsaurus* and *Yinlong* than in *Liaoceratops* because of the larger contact area. Therefore, the resorption degree of the functional tooth in *Chaoyangsaurus* and *Yinlong* is also larger than in *Liaoceratops*. Scale bars equal 5 mm (**A–C**) and 3 mm (**D–F**).

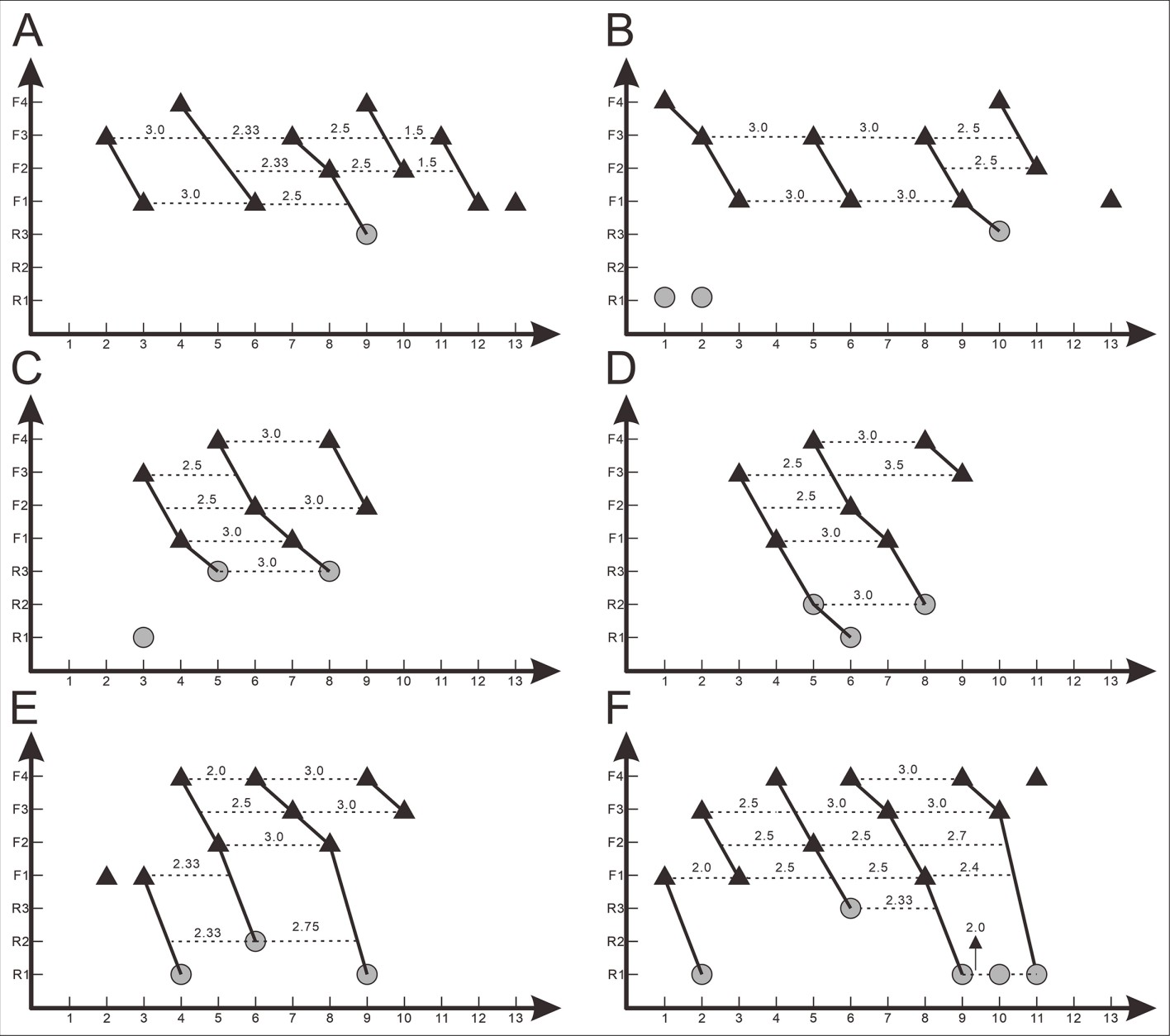

**Figure 8.** *Z*-Spacing diagrams of *Yinlong downsi* (IVPP V14530 and IVPP V18638) and *Chaoyangsaurus youngi* (IGCAGS V371). Zahnreihen graphs of right maxillary dentitions of IVPP V14530 (**A**) and right maxillary dentitions of IVPP V18638 (**B**). Zahnreihen graphs of IGCAGS V371 in the left maxilla (**C**), right maxilla (**D**), left dentary (**E**), and right dentary (**F**). The *x*-axis is the tooth position, *y*-axis is the tooth replacement stage. The black triangle represents the functional tooth and the gray circle represents the replacement tooth. Each imaginary line represents the *Z*-spacing which is the distance between Zahnreihen whose unit is a tooth position.

## Discussion

### Ontogenetic changes in dentitions of *Yinlong*

Although the accurate ontogenetic stage of these 4 specimens is not clear, the ontogenetic variation of the tooth replacement pattern in this taxon can be discussed relative to the specimens' size difference. Previous research suggests that the maxilla in *Y. downsi* bears 13 teeth (*Han et al., 2016*). Our 3D reconstructions reveal that 13 functional alveoli are preserved in the maxilla of V18638 and a larger individual (IVPP V14530). However, the count of functional teeth in the largest individual (IVPP V18637) is at least 14. Hence, the number of the maxillary teeth may increase with the ontogeny of *Y. downsi*, as in *Psittacosaurus mongoliensis* and *Protoceratops* (*Brown and Schlaikjer, 1940*; *Sereno et al.,*

*1990*; *Czepiński, 2019*). In large individuals (IVPP V18637, IVPP V14530), there is 1 replacement tooth out of 14 or 13 functional teeth in the maxilla whereas smaller specimens (IVPP V18637, IVPP V18638) have a higher ratio of the replacement teeth to the functional teeth such as 2 RT/8 FT and 3 RT/10 FT (*Table 1*). This phenomenon may reflect that the early ontogenetic stage specimens of *Yinlong* may have a faster tooth replacement rate. As noted by *He et al., 2018*, remnants of mostly resorbed functional teeth are present in both juvenile and adult specimens of *Liaoceratops*. But the remnants of resorbed functional teeth are only present in the largest specimen (IVPP V18637) of *Yinlong*. Therefore, we conclude that the resorption rate may decrease through the ontogeny of *Yinlong*.

## The evolution of dental anatomy and replacement pattern in Ceratopsia

Dental anatomy. *Tanoue et al., 2012* have concluded that the evolutionary trend in dentitions of early-diverging ceratopsians includes an increase in the angle of the wear facets, development of a prominent primary ridge, development of deep indentations on the mesial and distal sides of the primary ridge and increase in size in neoceratopsians. By computed tomographic analysis, we found that the dentitions in *Yinlong*, *Hualianceratops*, and *Chaoyangsaurus* exhibit features that differ from neoceratopsians including small numbers of teeth in tooth rows, concave surfaces on the lingual side of the maxillary crowns and labial side of the dentary crowns, loosely packed tooth rows, and regular occlusal surfaces. There are also some differences between early-diverging taxa. The crowns of unworn teeth in *Yinlong* and *Hualianceratops* are subtriangular and bear primary ridges located at the midline of the crowns (*Figures 1C and 6C*). Unlike *Yinlong* and *Hualianceratops*, the maxillary dentitions of *Chaoyangsaurus* developed ovate crowns and the relatively prominent primary ridge located relatively distal to the midline of the crowns as in most neoceratopsians (*Figure 5F, H*). In addition, the roots in *Yinlong* are straight, unlike *Chaoyangsaurus* whose functional roots are curved lingually (*Figure 9A–C*). Overall, the dentitions of *Yinlong* and *Hualianceratops* exhibit primitive conditions compared to *Chaoyangsaurus*.

*Psittacosaurus lujiatunensis* (IVPP V12617) exhibits similar concave surfaces on the occlusal surface of the crowns as in *Yinlong*, *Chaoyangsaurus*, and *Hualianceratops*. These early-diverging ceratopsians bear similar low-angled wear facets but the depression on the occlusal surface indicates a different occlusion from the shearing occlusal system as in neoceratopsians. In addition, the primary ridges are located at the midline of the crowns in *P. lujiatunensis*.

In *Liaoceratops*, *Archaeoceratops*, and *Auroraceratops*, the crowns developed slightly more prominent and narrow primary ridges and the teeth of *Leptoceratops* and *Protoceratops* developed the most prominent primary ridge outside of ceratopsids (*Sues et al., 2009*). Significantly, the primary ridges in the dentary teeth in *Archaeoceratops* (IVPP V11114) are located relatively mesial to the midline of the crowns in contrast to its maxillary dentitions and other neoceratopsians. Late-diverging neoceratopsians including *Leptoceratops* and *Protoceratops* have deeper indentations mesial and distal to the primary ridge, as in ceratopsids, than early-diverging neoceratopsians (*Sues et al., 2009*). In *Liaoceratops*, *Protoceratops*, *Leptoceratops*, and *Zuniceratops* which bear closer-packed dentitions, shallow longitudinal sulci form on the roots to accommodate adjacent crowns in neighboring tooth families. This allows for closer packing of the dentition (*Figure 9C, F*; *Brown and Schlaikjer, 1940*; *Wolfe et al., 1998*; *He et al., 2018*). Among all specimens we examined here, the occlusal surfaces of the functional teeth are regular and generally on the same plane whereas they are irregular in *Protoceratops* and Ceratopsidae (*Edmund, 1960*; *Sues et al., 2009*; *Mallon et al., 2016*). Differing from early-diverging ceratopsians, ceratopsids have evolved unique dental features including two-rooted teeth, high angle wear facets, and a very prominent primary ridge flanked by deep indentations (*Edmund, 1960*; *Sues et al., 2009*).

Replacement progression. The replacement progression in *Yinlong* and *Chaoyangsaurus* differs slightly from that of *Liaoceratops* (*He et al., 2018*). The resorption of the functional tooth in *Liaoceratops* is initiated after the replacement tooth grew, in contrast to *Yinlong* and *Chaoyangsaurus* (*He et al., 2018*). When the replacement tooth growth is nearly complete, the labial dentine of the roots in *Liaoceratops* remains more completely preserved than in *Yinlong* (*Figure 7B,C,E,F*). In addition, the root of the replacement tooth in *Liaoceratops* inclines lingually at 24° and that in *Yinlong* also inclines lingually but with a smaller angle of inclination (12°), and the root of the replacement tooth in *Chaoyangsaurus* is relatively vertical and is appressed to the functional tooth (*Figure 7A–C*). As a result,

**Table 1.** List of the ontogenetic difference in specimens of *Yinlong*.

| Specimen number | Alveoli | | | | | | The replacement teeth | | | | | | Resorbed functional teeth |
| | Premaxilla | | Maxilla | | Dentary | | Premaxilla | | Maxilla | | Dentary | | |
| | Left | Right | Left | Right | Left | Right | Left | Right | Left | Right | Left | Right | |
|---|---|---|---|---|---|---|---|---|---|---|---|---|---|
| IVPP V18638 | n.p. | n.p. | n.p. | 13 | n.p. | n.p. | n.p. | n.p. | n.p. | 3 | n.p. | n.p. | 0 |
| IVPP V18636 | 2 | 1 | 12* | 10* | 9* | 12* | 0 | 0 | 0 | 2 | 0 | 0 | 0 |
| IVPP V14530 | 3 | 3 | 13 | 13 | 15 | 14 | 0 | 0 | 1 | 1 | 1 | 1 | 0 |
| IVPP V18637 | 3 | 3 | 14 | 14 | n.p. | n.p. | 2 | 0 | 0 | 1 | n.p. | n.p. | Right maxilla: 2 |

n.p. = not preserved.

* represents the loss of alveoli.

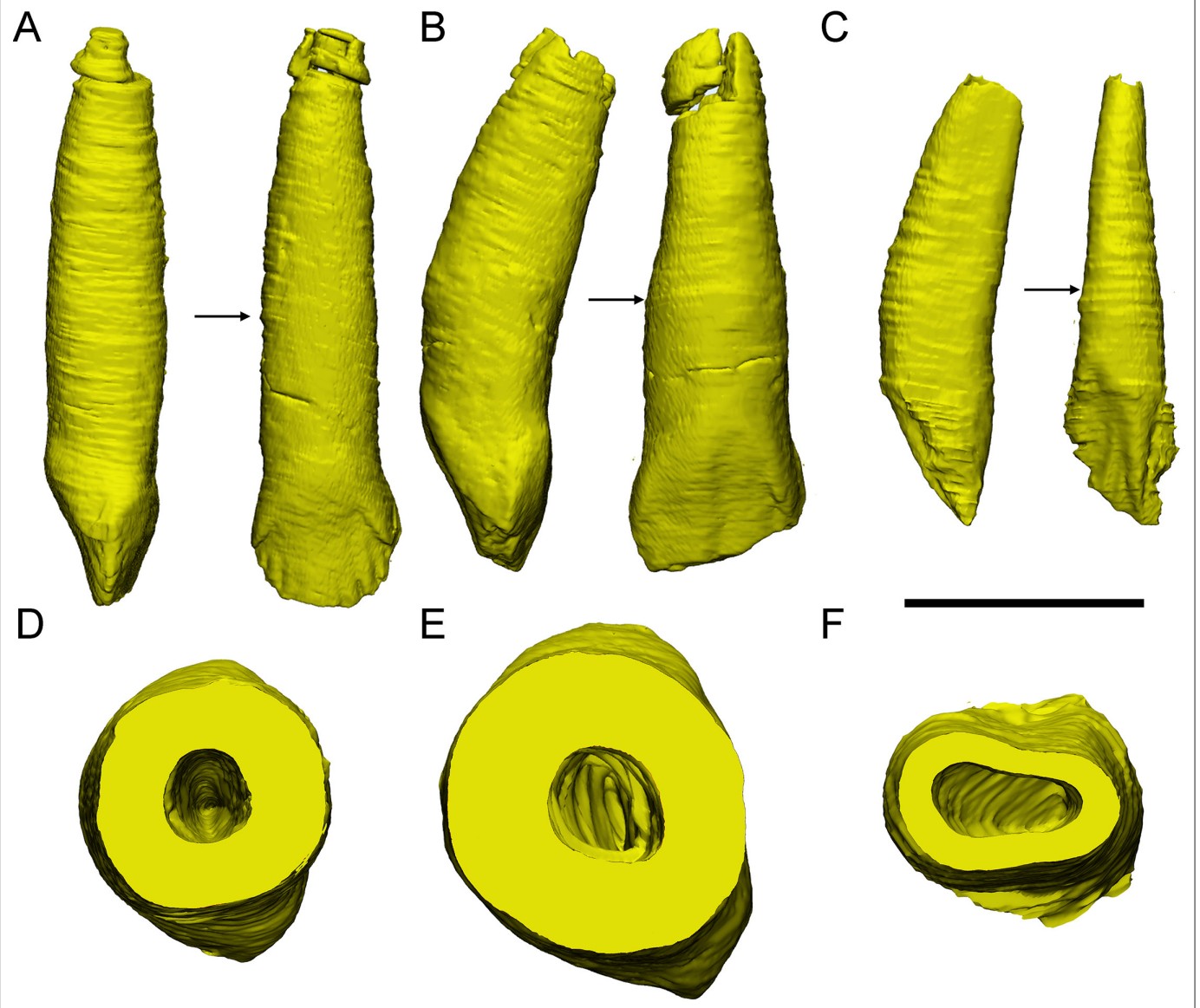

**Figure 9.** The reconstructions of three functional maxillary teeth at the middle part of the tooth row. The tooth 6 in IVPP V18638 in mesial, labial view (**A**), and cross-section (**D**). The tooth 7 in the left maxilla of IGCAGS V371 in mesial, labial view (**B**), and cross-section (**E**). The tooth 9 in the left maxilla of the holotype of *Liaoceratops* in distal, labial view (**C**), and cross-section (**F**). The arrows indicate where the cross-sections generate. Scale bar equals 10 mm (**A–C**) and 5 mm (**D–F**).

the far labial side of the root in *Liaoceratops* and *Yinlong* possibly lies beyond the zone of resorption and the dentine of the functional tooth next to the replacement tooth is still preserved, while that in *Chaoyangsaurus* is resorbed (*Figure 7*; *He et al., 2018*). In general, the degree of resorption of the functional tooth root is most severe in *Chaoyangsaurus* followed by *Yinlong*, and it is the weakest in *Liaoceratops*. In addition, the functional crown detaches from the root in *Liaoceratops* and the functional root remnants are still present labial to the replacement tooth while the functional tooth is shed (*He et al., 2018*). The relatively slight resorption and the separation between the resorbed functional crown and root may explain why remnants of the functional teeth are so prevalent in *Liaoceratops*.

At present, the replacement process in ceratopsids has not been described in detail. Some transverse sections previously reported suggested a difference in the replacement process between ceratopsids and early-diverging ceratopsians (*Erickson et al., 2015*). In ceratopsids, the replacement teeth germinated inside the pulp cavities of the predecessors instead of lingual to the root of predecessors (*Erickson et al., 2015*; *Figure 1B*). The transition of the location of the replacement teeth from the

lingual side of the functional roots to the tip of that has been reported in *Leptoceratops* (*Brown and Schlaikjer, 1940*) and may represent the primitive state in ceratopsids. This may explain the transition to double-rooted teeth in ceratopsids, where the replacement tooth is positioned between the labial and lingual roots of the functional tooth (*Erickson et al., 2015*; *Figure 1B*). As the teeth developed, the long axes of the replacement teeth in the same alveolus inclined from labially to lingually (*Erickson et al., 2015*; *Figure 1B*). The roots of the preceding functional teeth in ceratopsids would shed after the crowns have been worn away instead of mostly resorbed as they do in early-diverging ceratopsians (*Edmund, 1960*).

Tooth replacement pattern. Besides the morphological differences, a high rate of tooth replacement characterizes ceratopsids, identified by more replacement teeth in each vertical series (*Erickson, 1996*). In early-diverging neoceratopsians (*Liaoceratops* and *Auroraceratops*), an alveolus bears at most 2 replacement teeth with a relatively lower replacement rate (*Tanoue et al., 2012*; *He et al., 2018*; *Morschhauser et al., 2019*). In most early-diverging species of ceratopsians (*Yinlong*, *Chaoyangsaurus*, *Psittacosaurus*, and *Hualianceratops*), each alveolus bears at most 1 replacement tooth indicating lower replacement rates than late-diverging ceratopsians (*Table 2*).

Overall, the evolution of dentitions from the earliest-diverging ceratopsians to ceratopsids are as follows: the development of the primary ridges and the deep indentations; the increased angle of the wear facets on the crowns; the increase of tooth counts in tooth rows; the presence of the shallow grooves on the roots trending from single- to two-rooted teeth; the arrangement of teeth into a more compact mass; the increase of teeth in each tooth family; the location of the replacement teeth transferring from the lingual side of the functional teeth to the inside of the pulp cavities (*Figure 10*).

## Implications for diet and environment

The upper half of the Shishugou Formation, in which the bonebeds containing *Yinlong* and *Hualianceratops* occur, indicates a warm and seasonally dry climate in the Middle and Late Jurassic (*Eberth et al., 2001*; *Clark et al., 2004*; *Eberth et al., 2006*; *Bian et al., 2010*; *Eberth et al., 2010*). *Wang et al., 2000* have described the megaplant fossils *Equisetites* and *Elatocladus*, and pollen and spores of *Hymenophyllum*, *Anemia*, and *Cicatricosisporites*. This area developed forests near the banks of rivers under moist conditions and consisted primarily of conifers like Araucariaceae and the understory of the forest mainly consisted of *Angiopteris*, *Osmunda*, and *Coniopteris* (*McKnight et al., 1990*; *Hinz et al., 2010*). Feeding strategy can be inferred from its body size and tooth pattern. The holotype of *Yinlong* is estimated to be 120 cm in total body length (*Xu et al., 2006*), which implies that *Yinlong* likely feeds on low-growing plants such as *Equisetites*.

*Maiorino et al., 2018* pointed out that *Yinlong* was not able to tolerate high loadings due to its more primitive lower jaw morphology, and may have fed on softer foliage and fruits or swallowed the food in a relatively unprocessed form. In addition, although the tooth replacement rate in *Yinlong* is not clear, previous researchers have suggested that the tooth replacement rates in some sauropods and hadrosaurids, which have elaborate dental batteries, are relatively fast (*D'Emic et al., 2013*). The low number of replacement teeth in *Yinlong* likely reflects slow tooth replacement rates which would not imply rapid tooth wear. All these features suggest that *Yinlong* is unlikely to grind tough foods. Therefore, *Yinlong* possibly has food processing strategies other than grinding food with their dentitions. *Xu et al., 2006* noticed that the ribcage of IVPP V14530 preserved 7 gastroliths, which is also known in some other ornithischians (i.e., *Psittacosaurus Osborn, 1923*; *Cerda, 2008* and some non-avian theropods *Kobayashi et al., 1999*; *Fritz et al., 2016*). Furthermore, an armoured dinosaur *Borealopelta markmitchelli* with ingested stomach contents and gastroliths preserved has been reported recently and represents the most well-supported and detailed direct evidence of diet in a herbivorous dinosaur (*Brown et al., 2020*). The diet of *B. markmitchelli* includes selective ferns, preferential ingestion of leptosporangiate ferns, and incidental consumption of cycad-cycadophyte and conifer (*Brown et al., 2020*). *B. markmitchelli* possessed simple teeth and gastroliths and likely occupied similar ecological niches as *Yinlong*. In such a context, we suggest that ferns such as *Angiopteris*, *Osmunda*, and *Coniopteris* are suitable to be food choices of *Yinlong*. Some low and tender leaf and other less abrasive plant foods could also be possible. Early-diverging ceratopsians that show relatively slow tooth replacement rates and lack evidence of heavy tooth wear likely used gastroliths to triturate foodstuffs to cope with the stringent requirements for digestion of plant materials.

**Table 2.** List of the number of replacement teeth and the functional teeth in some ceratopsians which have been studied by computed tomography.

| Higher taxa | Genus | Specimen number | Left maxilla | | Right maxilla | | Left dentary | | Right dentary | |
|---|---|---|---|---|---|---|---|---|---|---|
| Ceratopsia | *Psittacosaurus* | CUGW VH104 | 9 FT | 7 RT | 9 FT | 5 RT | 9 FT | 6 RT | 10 FT | 7 RT |
| Ceratopsia | *Yinlong* | IVPP V14530 | 13 FT | 1 RT | 13 FT | 1 RT | 14 FT | 1 RT | 14 FT | 2 RT |
| Ceratopsia | *Chaoyangsaurus* | IGCAGS V371 | 9 FT | 3 RT | 9 FT | 3 RT | 9 FT | 3 RT | 11 FT | 5 RT |
| Neoceratopsia | *Liaoceratops* | IVPP V12738 | 13 FT | 11 RT and 1 2nd RT | 13 FT | 11 RT and 1 2nd RT | 15 FT | 13 RT and 2 2nd RT | 15 FT | 12 RT |
| Neoceratopsia | *Auroraceratops* | CUGW VH106 | - | - | - | - | - | - | 15 FT | 7 RT |

RT = replacement tooth; FT = functional tooth; 2nd RT = the second generation replacement tooth.

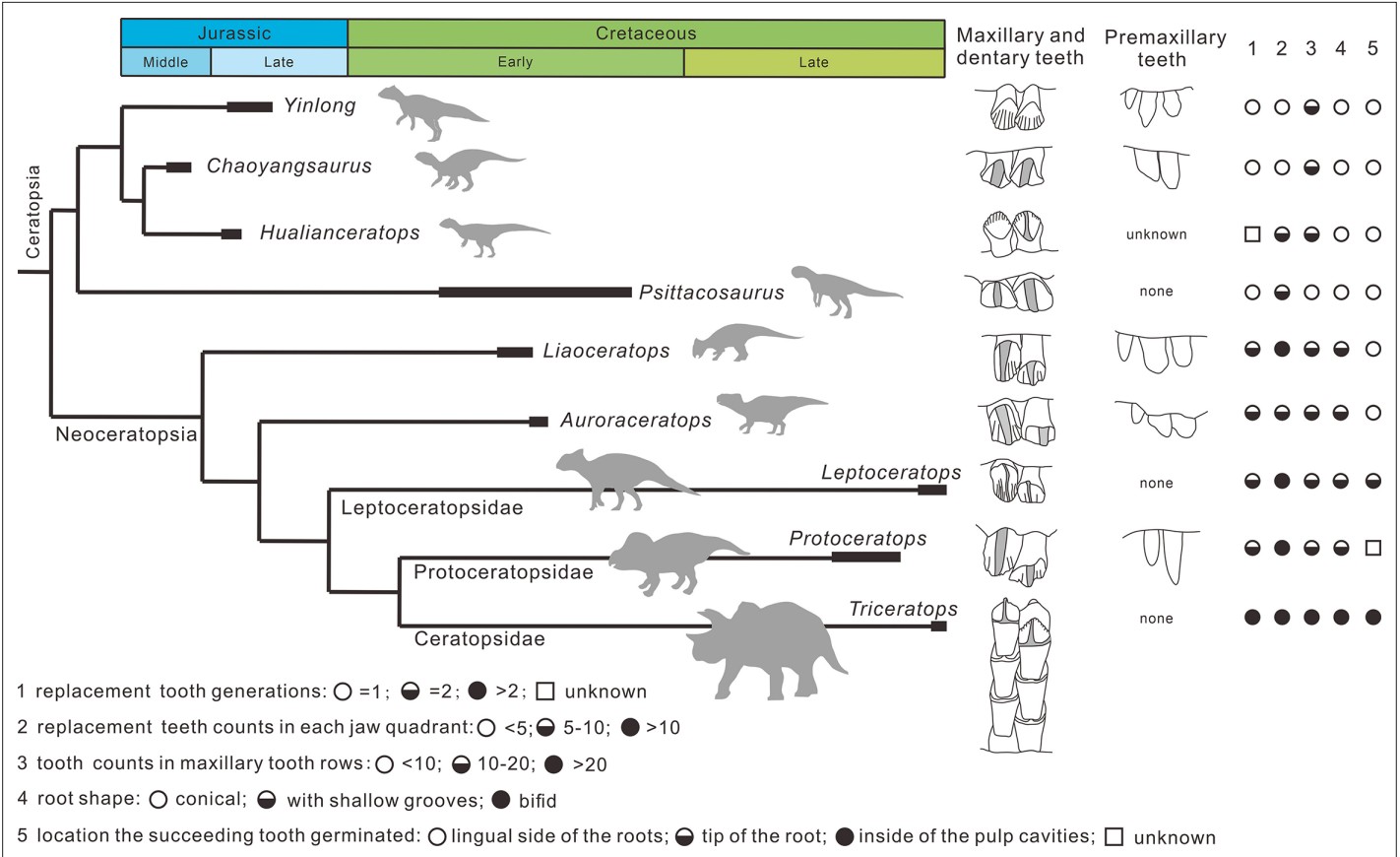

**Figure 10.** Phylogenetic tree of ceratopsians (composite from *Erickson et al., 2015*, *Han et al., 2018*, and *Yu et al., 2020*) and comparison of the dental anatomy and the tooth replacement pattern. *Psittacosaurus* from *Averianov et al., 2006*; *Liaoceratops* from *He et al., 2018*; *Auroraceratops* from *Tanoue et al., 2012* and *Morschhauser et al., 2019*; *Leptoceratops* from *Tanoue et al., 2012*; *Protoceratops* from *Edmund, 1960* and *Brown and Schlaikjer, 1940*; *Triceratops* from *Edmund, 1960*.

Several morphological adaptations occurred during the evolution of Ceratopsia including the longitudinal ridge of ceratopsids and thickening of the lower jaw in early-diverging neoceratopsians besides the transition of dentitions mentioned above (*Bell et al., 2009*; *Maiorino et al., 2018*). Finite element analysis on the lower jaws of ceratopsians suggests that ceratopsids represent the clade with the most efficient masticatory apparatus in Ceratopsia whereas the early-diverging ceratopsians *Hualianceratops* and *Yinlong* retained a primitive lower jaw (*Maiorino et al., 2018*). These changes undoubtedly improved the chewing ability in neoceratopsians and ceratopsids. Given their body difference, the greater food consumption brought by the increased body size may have driven, in part, the evolution of the jaw and the replacement patterns. However, increased body size may not be the only reason for increased replacement tooth number and the stronger jaw *Liaoceratops* and *Psittacosaurus* are similar in size to *Yinlong* but have more replacement teeth than *Yinlong* as well as 2 generations of replacement teeth in *Liaoceratops* and the jaws able to withstand higher stress (*He et al., 2018*; *Maiorino et al., 2018*). The Jehol flora, which occurs in the Yixian Formation of Liaoning, is dominated by Cycadopsida and Coniferopsida (*Deng et al., 2012*). It suggests that *Liaoceratops* had a different diet strategy from *Yinlong*. Likewise, one of the greatest changes in terrestrial ecosystems during the Late Cretaceous Period saw the diversification of angiosperms (*Barrett and Willis, 2001*). Changes in the floral composition may have resulted in the different diet strategies in ceratopsids which in turn may help explain the different tooth replacement patterns and rates.

## Materials and methods

### Institutional abbreviations

IVPP – Institute of Vertebrate Paleontology and Paleoanthropology, Beijing, China; IGCAGS – Institute of Geology Chinese Academy of Geosciences, Beijing, China.

### Material

Three earliest-diverging ceratopsians *Yinlong*, *Chaoyangsaurus*, and *Hualianceratops* were examined. The skull and mandible materials of *Yinlong* have been described in detail previously (*Han et al., 2016*; *Han et al., 2018*). Four skulls of *Y. downsi* are included in this study, IVPP V14530 (the holotype), IVPP V18636, IVPP V18637, and IVPP V18638.

IVPP V18638 (CT scanned). This is the smallest specimen of *Yinlong* described here with the skull length (measured from the rostral end to the posterior surface of the quadrate condyles) estimated to be about 13 cm. Only the right maxilla, jugal, squamosal, postorbital, quadratojugal, and pterygoid are preserved (*Figure 1*). The right maxillary dentition was reconstructed.

IVPP V18636 (CT scanned). This specimen consists of a nearly complete skull with a mandible and partial postcranial skeleton (*Figure 2*). The skull length is about 15.5 cm. The dentitions of the premaxillae, the maxillae, and the dentary are reconstructed.

IVPP V14530 (CT scanned). The holotype preserves a nearly complete skull with a mandible and nearly complete postcranial skeleton (*Figure 3*). The skull length is about 18 cm. The dentitions of the premaxillae, the maxillae, and the dentary are reconstructed.

IVPP V18637 (CT scanned). The preserved elements on this specimen consist of a nearly complete skull lacking a mandible (*Figure 4*). It is the largest specimen of *Yinlong* with a skull length measured as 23 cm. Only the premaxillary and maxillary dentitions are studied.

The holotype of *Chaoyangsaurus* (IGCAGS V371) includes the dorsal part of a skull and a nearly complete mandible (*Figure 5*; *Zhao et al., 1999*). The skull and the mandible of IGCAGS V371 were CT scanned, respectively. The dentitions of the premaxillae, the maxillae, and the dentary are reconstructed.

The holotype of *Hualianceratops* (IVPP V18641) was also CT scanned, but we were unable to study teeth due to poor preservation. An additional specimen, IVPP V28614 (field number WCW-05A-2), which only preserves the left dentary, is described here for comparison although it was not CT scanned (*Figure 6*). However, the external morphology provided information on the tooth replacement pattern. We assigned this specimen to *Hualianceratops* based on the deep and short dentary which measures 83.46 mm in length and has a depth of 33.38 mm at the rostral end (40% length) and strongly rugose sculpturing present on the lateral surface of the dentary (*Figure 6A*; *Han et al., 2015*).

Each functional tooth and replacement tooth's total height, maximum mesiodistal width, and maximum labiolingual width of all studied specimens are displayed in *Supplementary file 1*.

### Computed tomography

The roots of the functional teeth and the replacement teeth are usually encased in the tooth-bearing elements. By employing traditional methods, it is difficult to obtain the internal anatomical features of

**Table 3.** Skull length and scanning parameters of *Yinlong* and *Chaoyangsaurus*.

| Taxa | Specimen number | Skull length* (cm) | Scanning voltage | Scanning current | Resolution (µm) |
|------|-----------------|-----|------------------|------------------|-----------------|
| *Yinlong* | IVPP V18638 | 13.4 (uncomplete) | 130 kV | 140 mA | 36.039 |
| *Yinlong* | IVPP V18636 | 15.5 | 430 kV | 1500 µA | 160 |
| *Yinlong* | IVPP V14530 | 18 | 430 kV | 1500 µA | 300 |
| *Yinlong* | IVPP V18637 | 23 | 430 kV | 1500 µA | 160 |
| *Chaoyangsaurus* | IGVAGS V371 | 13.7 | 150 kV | 160 mA | 46.493 |

*Skull length is measured from the rostral end to the posterior surface of the quadrate condyles.

the dentitions in any detail. The advent of noninvasive and nondestructive radiological approaches, X-ray computed tomography, has revolutionized the study of fossil specimens (*Conroy and Vannier, 1984*), providing new insights into internal structures normally obscured by bones and rock matrix. Here, high-resolution X-ray micro-computed tomography was used to reveal internal anatomical features of teeth and tooth replacements in the premaxillae, maxillae, and dentary. Scanning of IVPP V14530, IVPP V18636, and IVPP V18637 was carried out using a 450 kV micro-computed tomography instrument (450 ICT) at the Key Laboratory of Vertebrate Evolution and Human Origins of the Chinese Academy of Sciences, Beijing, China. Scanning on IVPP V18638 and IGCAGS V371 was carried out using a 300 kV micro-computed tomography instrument (Phoenix Vtomex M) and the detector (Dynamic41-100) at the Key Laboratory of Vertebrate Evolution and Human Origins of the Chinese Academy of Sciences, Beijing, China. Scanning parameters of these specimens are displayed in *Table 3*. High-resolution 3D models of the dentitions of *Yinlong* and *Chaoyangsaurus* are available in Dryad, at https://doi.org/10.5061/dryad.9ghx3ffk0.

CT datasets were input in Mimics (Materialise Corporation, Leuven, Belgium, versions 15.0 and 16.0) to render 3D models of bones and teeth. The program builds meshes based on density differences in each specimen and applies material properties to each mesh.

## The reconstruction of Zahnreihen

*Edmund, 1960* hypothesized that teeth in reptiles are replaced in an ordered, alternating segmented pattern called a Zahnreihe. Each Zahnreihe consists of a series of teeth, including unerupted teeth, where a rostrally placed tooth is more mature than a more caudal one along the same tooth row (*Hanai and Tsuihiji, 2019*). The distance between two successive Zahnreihen is the Z-spacing (*DeMar, 1972*). Previous researchers usually defined the Zahnreihen by measurements of teeth or by applying a replacement index (*Demar and Bolt, 1981*; *Fastnacht, 2008*; *He et al., 2018*). Because few replacement teeth are preserved in *Yinlong* and *Chaoyangsaurus*, it is difficult to reconstruct the tooth replacement waves by applying the same replacement index used in *Liaoceratops*. Therefore, we reconstructed the Zahnreihen according to the degree of tooth wear and the location of replacement teeth, as used in *Shunosaurus* (*Chatterjee and Zheng, 2002*), as well as applying a new methodology that includes the developmental stage of the pulp cavity. We divided the functional teeth in *Yinlong* and *Chaoyangsaurus* into 4 stages: (F1) no or slight wear on marginal denticles with an open pulp cavity; (F2) wear on marginal denticles and a slightly concave lingual wear facet with a large pulp cavity; (F3) extensive wear on marginal denticles and a concave lingual wear facet with the depression on the lingual surface of the roots or a bud of the replacement tooth; (F4) polished and greatly worn marginal denticles and a highly concave lingual wear facet with a broken pulp cavity or the emergence of a replacement tooth. Three stages of replacement teeth are recognized: (R1) small incipient tooth showing the tip of the crown; (R2) crown fully erupted; (R3) crown reaches the base of the functional crown.

Based on stage division, each functional tooth and replacement tooth was plotted on a graph whose vertical axis is the growth stage and the horizontal axis is the tooth position. In the graph, these teeth show a regular pattern that the growth stage decreases progressively and periodically over several tooth positions. Each degressive sequence represents a Zahnreihe indicated by a series of teeth linked with each other as black lines (*Figure 8*). The distance between adjacent Zahnreihen is Z-spacing and the Z-spacing of *Yinlong* is described by the mean of all measurements.

## Acknowledgements

We thank the members of the Sino-American expedition team for collecting the fossils described herein, and L S Xiang, T Yu, and X Q Ding for preparing the fossils. Yun Feng and Y M Luo for helping CT scan. Yang Wu and Yuzheng Ke for helping reconstruct CT models. Yonatan Sahle, Marcos Gabriel Becerra, and an anonymous referee for their comments. This project was supported by the National Natural Science Foundation of China and the International Partnership Program of Chinese Academy of Sciences.

## Additional information

### Funding

| Funder | Grant reference number | Author |
|---|---|---|
| National Natural Science Foundation of China | 41972021 | Fenglu Han |
| National Natural Science Foundation of China | 42288201 and 41688103 | Xing Xu |
| National Natural Science Foundation of China | 42072008 | Qi Zhao |
| International Partnership Program of Chinese Academy of Sciences | 132311KYSB20180016 | Xing Xu |

The funders had no role in study design, data collection, and interpretation, or the decision to submit the work for publication.

### Author contributions

Jinfeng Hu, Investigation, Three-dimensional models reconstruction; Figures production, Visualization, Writing - original draft; Catherine A Forster, Conceptualization, Data curation, Validation, Writing – review and editing; Xing Xu, Conceptualization, Supervision, Writing – review and editing; Qi Zhao, Data acquisition and interpretation, Data acquisition and interpretation, Data curation, Writing – review and editing; Yiming He, Data acquisition and interpretation, Data acquisition and interpretation, Data curation, Writing – review and editing; Fenglu Han, Conceptualization, Funding acquisition, Methodology, Supervision, Writing – review and editing

### Author ORCIDs

Jinfeng Hu http://orcid.org/0000-0001-9237-9756
Xing Xu http://orcid.org/0000-0002-4786-9948
Fenglu Han http://orcid.org/0000-0003-3399-4008

### Decision letter and Author response

Decision letter https://doi.org/10.7554/eLife.76676.sa1
Author response https://doi.org/10.7554/eLife.76676.sa2

## Additional files

### Supplementary files

• Supplementary file 1. List of each functional and replacement tooth's total height, maximum mesiodistal width, maximum labiolingual width, and the height of tooth remnants of all specimens.

• Transparent reporting form

### Data availability

All data generated or analyzed during this study are included in the manuscript and supplementary file. We have uploaded the raw micro-CT scanning images of all scanned specimens (all cropped to the dentigerous regions) in Dryad as .TIF or .BMP file format and also the reconstructed 3D files (see the link https://doi.org/10.5061/dryad.9ghx3ffk0). The detailed information of all images is provided in a TXT file 'README_file.txt' saved in Dryad.

The following dataset was generated:

| Author(s) | Year | Dataset title | Dataset URL | Database and Identifier |
|---|---|---|---|---|
| Jinfeng H | 2022 | Data from: Computed tomographic analysis of dental system of three Jurassic ceratopsians: implications for the evolution of the tooth replacement pattern and diet in early-diverging ceratopsians | https://dx.doi.org/10.5061/dryad.9ghx3ffk0 | Dryad Digital Repository, 10.5061/dryad.9ghx3ffk0 |

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
