## [Editor Report]

This paper employs virtual analytical methods to thoroughly examine the evolution of the dental system of Jurassic ceratopsian dinosaurs. The rich dataset and level of analytical detail on tooth development and replacement patterns are accompanied by careful interpretations of functional morphology and dietary adaptation, making the paper particularly valuable for main-stream paleontologists while still appealing to the wider evolutionary biology community.

---

## [Decision Letter]

**Decision letter after peer review:**

Thank you for submitting your article "Computed tomographic analysis of the dental system of three Jurassic ceratopsians: implications for the evolution of the tooth replacement pattern and diet in early-diverging ceratopsians" for consideration by *eLife*. Your article has been reviewed by 3 peer reviewers, including Yonatan Sahle as the Reviewing Editor and Reviewer #1, and the evaluation has been overseen by George Perry as the Senior Editor. The following individual involved in review of your submission has agreed to reveal their identity: Marcos Gabriel Becerra (Reviewer #2).

Essential revisions:

1) To demonstrate that the conclusions in the article are supported by the data presented, the authors should add the raw metric data on the dimensions of each functional and replacement tooth.

2) Raw micro-CT scans (at least ones cropped to the dentigerous regions only but preserve scaling information) must be made available via a suitable permanent archive. If this is not possible for any reason, the authors must discuss their justification with the Senior Editor.

*Reviewer #1 (Recommendations for the authors):*

Abstract

The last few sentences of the abstract focus exclusively on Yinlong and can benefit from a sentence on new information about the two other Ceratopsian genera that make up the focus of the paper.

Materials and Methods

The first paragraph can benefit from a revision in which only a quick summary of the type, size and preservation details of each studied specimen is presented, with the method of analysis saved for the respective subtitles. Also, a clearer mention how much dentition is preserved and studied for the respective genera will be helpful.

The subtopic on micro-CT methods could use why the method was chosen over other approaches, especially since this is also reflected in the title of the paper.

Line 90: Add the location of the laboratory (i.e., Beijing, China).

Lines 99-100: Be consistent with the use of "Zahnreihen" as the plural form of "Zahnreihe"

Line 100: Previous researchers…

Line 354: Replace "their" with "the specimens'" for clarity

Line 453: Consider revising subtitle as "Implications for…"

*Reviewer #2 (Recommendations for the authors):*

The manuscript denotes not only excellent research but also effort and time well spent to perform the 3D reconstructions. The use of opened/closed pulp cavity and its relation with wear development is innovative evidence that supports the interpretations of stages of tooth replacement, and the discussion of the specific stage represented by Yinlong, Hualianceratops, and Chaoyangsaurus in the phylogenetic framework of ceratopsian dinosaurs opens the discussion of the similarities and differences with other ornithischian lineages that also evolved specialized herbivory.

*Reviewer #3 (Recommendations for the authors):*

Could you estimate tooth formation times and replacement rates based on tooth length?

---

## [Author Response]

Essential revisions:1) To demonstrate that the conclusions in the article are supported by the data presented, the authors should add the raw metric data on the dimensions of each functional and replacement tooth.

Thank you for pointing this out. We have provided each functional and replacement tooth’s total height, maximum mesiodistal width, maximum labiolingual width of all specimens presented in TABLE S1 classified by the specimen number. These data help to support our conclusions.

2) Raw micro-CT scans (at least ones cropped to the dentigerous regions only but preserve scaling information) must be made available via a suitable permanent archive. If this is not possible for any reason, the authors must discuss their justification with the Senior Editor.

We have uploaded the raw micro-CT scanning images of all scanned specimens (all cropped to the dentigerous regions) in Dryad as.TIF or.BMP file format and also the reconstructed 3D files (see the link https://doi.org/10.5061/dryad.9ghx3ffk0). The detailed information of all images is provided in a TXT file ‘README_file.txt’ saved in Dryad.

Reviewer #1 (Recommendations for the authors):AbstractThe last few sentences of the abstract focus exclusively on Yinlong and can benefit from a sentence on new information about the two other Ceratopsian genera that make up the focus of the paper.

Thanks for your suggestion. We have added more statements on the teeth replacement pattern of *Chaoyangsaurus* and *Hualianceratops* in the abstract (Page 2, Line 44-49).

Materials and MethodsThe first paragraph can benefit from a revision in which only a quick summary of the type, size and preservation details of each studied specimen is presented, with the method of analysis saved for the respective subtitles. Also, a clearer mention how much dentition is preserved and studied for the respective genera will be helpful.

Thanks for your suggestion. We have reorganized the materials and the methods and made a summary of each specimen with their preservation details, size, and preserved dentitions (Page 29-30, Line 656-685).

The subtopic on micro-CT methods could use why the method was chosen over other approaches, especially since this is also reflected in the title of the paper.

Considering the reviewer’s suggestion, we have supplemented the advantage of the micro-CT methods in the first paragraph of the computed tomography. The reasons for using the method are as follows: The main body of the functional teeth and the replacement teeth are usually encased in the tooth-bearing bones excluding the crowns. Micro-CT methods help us to study the internal structures and the tooth replacement progress through the bone and rock matrix (Page 30, Line 694-698).

Line 90: Add the location of the laboratory (i.e., Beijing, China).

Added as suggested (Page 31, Line 703, 707).

Lines 99-100: Be consistent with the use of "Zahnreihen" as the plural form of "Zahnreihe"

Corrected (Page 31, Line 723).

Line 100: Previous researchers…

Modified as suggested (Page 31, Line 723).

Line 354: Replace "their" with "the specimens'" for clarity

Modified as suggested (Page 19, Line 443-444).

Line 453: Consider revising subtitle as "Implications for…"

Modified as suggested (Page 25, Line 579).

Reviewer #3 (Recommendations for the authors):Could you estimate tooth formation times and replacement rates based on tooth length?

Tooth formation times and replacement rate can be calculated in extinct animals by counting incremental lines of deposition in tooth dentin. However, more histological experiments need to be done to acquire the mean width of von Ebner incremental lines. These experiments require not only a large amount of time but also the permission of the museum. This will be our research focus of the next stage. The current focus of our study is the tooth replacement pattern of earliest-diverging ceratopsians.